# Structure Based Affinity Maturation and Characterizing of SARS-CoV Antibody CR3022 against SARS-CoV-2 by Computational and Experimental Approaches

**DOI:** 10.3390/v14020186

**Published:** 2022-01-19

**Authors:** Wei Yu, Nan Zhong, Xin Li, Jiayi Ren, Yueming Wang, Chengming Li, Gui Yao, Rui Zhu, Xiaoli Wang, Zhenxing Jia, Changwen Wu, Rongfeng Chen, Weihong Zheng, Huaxin Liao, Xiaomin Wu, Xiaohui Yuan

**Affiliations:** 1Institute of Biomedicine, Jinan University, Guangzhou 510632, China; wei926yu@163.com (W.Y.); jeasons@163.com (N.Z.); linklixin@163.com (X.L.); jiayi80@126.com (J.R.); ws_wym@126.com (Y.W.); cli816@163.com (C.L.); 2Anhui Province Key Laboratory of Pollutant Sensitive Materials and Environmental Remediation, College of Life Sciences, Huaibei Normal University, Huaibei 235000, China; 12007110659@chnu.edu.cn (G.Y.); njzr526@163.com (R.Z.); 3Guangdong Provincial Key Laboratory of Bioengineering Medicine, Guangzhou 510632, China; 4National Engineering Research Center of Genetic Medicine, Guangzhou 510632, China; 5School of Health, Zhuhai College of Science and Technology, Zhuhai 519041, China; 6Zhuhai Trinomab Biotechnology Co., Ltd., Zhuhai 519040, China; bolinjiari@163.com (X.W.); jiazhenxing@trinomab.com (Z.J.); swcw965@163.com (C.W.); chenrongfeng2013@163.com (R.C.); zwh13901154411@139.com (W.Z.)

**Keywords:** COVID-19, SARS-CoV-2, receptor-binding domain (RBD), CR3022, molecular dynamics (MD) simulation

## Abstract

The COVID-19 epidemic is raging around the world. Neutralizing antibodies are powerful tools for the prevention and treatment of SARS-CoV-2 infection. Antibody CR3022, a SARS-CoV neutralizing antibody, was found to cross-react with SARS-CoV-2, but its affinity was lower than that of its binding with SARS-CoV, which greatly limited the further development of CR3022 against SARS-CoV-2. Therefore, it is necessary to improve its affinity to SARS-CoV-2 in vitro. In this study, the structure-based molecular simulations were utilized to virtually mutate the possible key residues in the complementarity-determining regions (CDRs) of the CR3022 antibody. According to the criteria of mutation energy, the mutation sites that have the potential to impact the antibody affinity were then selected. Then optimized CR3022 mutants with the enhanced affinity were further identified and verified by enzyme-linked immunosorbent assay (ELISA), surface plasma resonance (SPR) and autoimmune reactivity experiments. Finally, molecular dynamics (MD) simulation and binding free energy calculation (MM/PBSA) were performed on the wild-type CR3022 and its two double-site mutants to understand in more detail the contribution of these sites to the higher affinity. It was found that the binding affinity of the CR3022 antibody could be significantly enhanced more than ten times after the introduction of the S103F/Y mutation in HCDR–3 and the S33R mutation in LCDR–1. The additional hydrogen-bonding, hydrophobic interactions, as well as salt-bridges formed between the modified double-site mutated antibody and SARS-CoV-2 RBD were identified. The computational and experimental results clearly demonstrated that the affinity of the modified antibody has been greatly enhanced. This study indicates that CR3022 as a neutralizing antibody recognizing the conserved region of RBD against SARS-CoV with cross-reactivity with SARS-CoV-2, a different member in a large family of coronaviruses, could be improved by the computational and experimental approaches which provided insights for developing antibody drugs against SARS-CoV-2.

## 1. Introduction

A global outbreak of novel coronavirus disease 2019 (Corona Virus Disease 2019, COVID-19) caused by severe acute respiratory syndrome coronavirus 2 (SARS-CoV-2) infection is currently spreading globally on a large scale [1,2]. In particular, the continuing emergence of new SARS-CoV-2 mutants poses a significant threat to global public health, human security and economic stability [3]. Promoting the development of drugs targeting the conserved region is crucial for responding to the high-frequency virus mutation and preventing and controlling the current pandemic [4]. SARS-CoV-2 is identified as the seventh coronavirus infecting humans [5]. Its spike protein (S) mediates the invasion of the host cells by binding the cell surface receptor, angiotensin converting enzyme 2 (ACE2) [6,7], and it is related to its host cell tropism [8,9]. Therefore, SARS-CoV-2 S protein, especially the receptor-binding domain (RBD) in the S1 subunit, is the primary target by neutralizing antibodies. The RBD sequence is composed of 193 amino acids with strong immunogenicity, which has been regarded as a critically important target for developing of therapeutics and vaccines [10].

It has been shown that the genomic similarity between SARS-CoV-2 and SARS-CoV reaches approximately 80% [10]. The two viruses are named SARS-related coronavirus and belong to the same species in the coronavirus family [10]. A great number of antibodies with potent neutralizing activity against SARS-CoV (such as M396, 80R, CR3022, 4A8) have been identified [11,12,13]. The M396/80R/4A8 antibodies exhibit lower affinities for their respective epitopes of SARS-CoV-2, while antibody CR3022 still shows certain binding to SARS-CoV-2 S protein [14]. After the outbreak of COVID-19, the crystal structure of CR3022–SARS-CoV-2 RBD (PDB: 6W41) was obtained by Yuan et al. [15]. Because of its cross neutralizing activity, CR3022 has the potential to be developed as a broadly neutralizing antibody against continuously mutating SARS-CoV-2 viruses. However, CR3022 exhibits much stronger binding activity to SARS-CoV RBD than to SARS-CoV-2 RBD [11]. The affinity enhancement of CR3022 could improve neutralizing activity against SARS-CoV-2 in vitro and in vivo. Thus, the affinity-enhanced CR3022 antibody could be much more attractive for being developed as a therapeutic for SARS-CoV-2 infection.

The interaction mode and key residue pairs of CR3022 with SARS-CoV RBD and SARS-CoV-2 RBD, as well as the cross-reaction of CR3022 between SARS-CoV and SARS-CoV-2, have been investigated by the previous studies [16,17]. It has been demonstrated that CR3022 binds to the conserved region away from the ACE2 receptor binding site on the RBDs of SARS-CoV and SARS-CoV-2 [15]. Thus, it can theoretically bind multiple coronavirus variants because most of the mutations of SARS-CoV-2 occurred in the RBD region that directly interacts with ACE2 [3,11]. Since the fall of 2020, multiple SARS-CoV-2 mutant strains have emerged and they are spreading unabated across the world [1]. For example, the South Africa variant, also known as B.1.351 [18], has led to the failure of a number of antibodies directed to the RBD receptor-binding site including the LYCoV-555 that has been initially granted and recently revoked for Emergency Use Authorization by the US FDA [19,20]. It has been anticipated and the evidence has been accumulated that SARS-CoV-2 will continuously mutate [6]. Therefore, neutralizing antibodies targeting a conserved region, such as CR3022 have a greater advantage than antibodies directly binding to the receptor-binding site with frequent mutation to be developed as a broad-spectrum antibody drug against the SARS-CoV-2 viruses.

In this study, computational virtual mutation (VM) [21] was systematically performed to simulate the CDRs high frequency mutation process of antibody in vivo, by saturation mutations of residues in the CDRs of CR3022 that are involved in the binding SARS-CoV-2 RBD (Figure 1). The possible hotspot residues on the interface of the CR3022 variant with improved affinity were identified by our computational system. The CR3022 variants were then expressed recombinantly in vitro and evaluated for the binding to SARS-CoV and SARS-CoV-2 RBD proteins by the enzyme-linked immunosorbent assay (ELISA) and surface plasma resonance (SPR) assays. Autoimmune reactivity was tested to determine the safety of CR3022 variants. Finally, we used molecular dynamics(MD) simulation and per-residue binding free energy calculation (MM/PBSA) to elucidate the molecular mechanism of the enhancement in the affinity of CR3022 mutants. This study provides a theoretical basis for the molecular design for developing broadly cross-reactive antibodies with a much higher affinity against SARS-CoV-2 infection.

## 2. Results and Discussion

### 2.1. Computational Virtual Mutation

The effect of mutations on amino acid residues in the CDRs of CR3022 on the binding SARS-CoV-2 RBD was evaluated by computational virtual saturation mutation. Virtual mutation (VM) was performed using the Binding-Mutation-Energy module of the Discovery Studio 4.5 software package. For every single mutant, the differences in the energy change (mutation energy) of binding between the wild type and mutated structures were calculated. Virtual mutations have been performed on two systems of CR3022(Fab) that corresponded to the crystal structure with PDB_ID 6W41 and the global optimized structure of N76200 extracted from the MD trajectory, respectively. The CR3022 mutants with favorable affinities were obtained with the mutation energy (Δ*G_bind_*) less than −1 kJ/mol as the screening criterion which reflected that the mutation exhibited higher binding stability and binding activity [22,23,24]. All virtual mutation calculations were performed on the amino acids in the CDR regions of CR3022. We have identified 18 mutation sites that were met the criteria for improvement, including eight single-site mutations in the variable region of the heavy chain (V_H_), and 10 single-site mutations in the variable region of the light chain (V_L_) were identified and listed in Table 1.

It was found that when the S103 residue on the heavy chain CDR3 was mutated to five different amino acids, the stability of antigen-antibody binding could be enhanced. The same was true when the S33 residue on the light chain CDR1 was mutated to nine different amino acids.

In order to explore the theoretical feasibility mentioned in this article, the synergistic effect brought by multiple mutations and the internal interaction between the sites need to be considered at the same time. On the other hand, considering the complexity of antigen-antibody interactions, their binding interface might involve various non-bonded interactions formed by multiple residues. It was unlikely that changing only a single residue in a tremendous protein system will improve the binding affinity of the antibody. Based on the results obtained by computational virtual results of single-point mutations, multi-site mutations on the CR3022 antibody were needed to further optimize the antigen-antibody interaction interface.

We chose two versions of mutants each with double mutations including S103F(H)-S33R(L) and S103Y(H)-S33R(L). The main reason for choosing S103 on the heavy chain (V_H_) was that the amino acids in the heavy chain CDR3 region of an antibody usually played an important role in the process of binding to the antigen. The reason for choosing S33 on the light chain (V_L_) was that the S33 site may be the main factor hindering the binding of the antibody to SARS-CoV-2. In addition, the other site, N35 on the light chain CDR1, was relatively close to the S33 site in space, so these sites were not suitable for simultaneous mutation. On the other hand, these two sites Heavy-S103 and Light-S33 were located far away from each other both in the primary sequence and in the tertiary structure. It did not give rise to steric hindrance and affect the structural stability of CR3022. The filtered mutation sites were subsequently expressed and purified for following experimental testing.

### 2.2. Binding of the Wild-Type and Variants of CR3022 to SARS-CoV-2 RBD and SARS-CoV RBD

Recombinant CR3022 antibody and variants with single amino acid substitutions and two double amino acid substitutions in the HCDR3 or LCDR1 were produced in 293 cells by transient transfection, purified by Protein A column chromatography. Analysis by SDS-PAGE under both non-reduced and reduced conditions indicated that the purified recombinant CR3022 antibody and variants all have a similar profile and purity. The integrity and biological activity of the single-site mutated and double-site mutated antibodies were kept intact. The light- and heavy- chains of the modified antibodies remained intact in SDS-PAGE electrophoresis analysis (Figure 2).

The binding of the wild-type and variants of CR3022 antibody to the SARS-CoV-2 RBD and SARS-CoV RBD were respectively evaluated by ELISA assays. As shown in Figure 3, the wild-type CR3022 antibody and its single-site mutants could bind tightly to the SARS-CoV RBD and SARS-CoV-2 RBD. Herein, the ELISA results obviously showed that the single-site modification of the CR3022 antibody could not affect its biological function of binding to the SARS-CoV RBD and SARS-CoV-2 RBD, despite the binding affinity of three mutants (S33F, S33W and S33L) to SARS-CoV-2 RBD being slightly reduced. Similarly, the biological activity of the double-site modification of the CR3022 antibody has not been changed either.

### 2.3. Binding Affinity Determined by SPR Analysis

The binding affinity of SARS-CoV RBD and SARS-CoV-2 RBD to the wild-type CR3022 and its 18 mutated antibodies was respectively measured by surface plasmon resonance (SPR) with GE Biacore 8K. We found that the wild-type CR3022 exhibited high affinity for SARS-CoV RBD with the slow association rate at 2.44 × 10^5^ M^−1^s^−1^ and dissociation at 5.27 × 10^−4^ s^−1^ with KD of 2.16 × 10^−9^ M. However, wild-type CR3022 bound SARS-CoV-2 RBD with much weaker binding affinity 1.49 × 10^−8^ M with faster association 1.17× 10^6^ M^−1^s^−1^ but much faster dissociation 1.74 × 10^−2^ s^−1^ (Figure 4), which is consistent with our previous work and other reports [15,16].

It also included a series of mutated antibodies in vitro (Appendix A). The binding kinetics curves showed that both SARS-CoV RBD and SARS-CoV-2 RBD could bind quickly and also rapidly disassociate from the antibody that was conjugated to a protein chip. Not all of the modified antibodies that had been screened successfully by computational virtual calculations were found to be consistent with the experimental results. Compared with the wild-type CR3022, the binding affinity of three modified antibodies (CR3022–S103F(H), CR3022–S103Y(H) and CR3022–S33R(L)) was found to be slightly improved and their dissociation curves displayed a slowing tendency, while the other mutated antibodies remained almost unchanged (Appendix A).

CR3022 was originally found to be bound to SARS-CoV, with an affinity constant of 2.16 nM (Figure 4A). As the affinity constant is shown in Figure 4B,C, the affinity constant of both modified antibodies to SARS-CoV-2 RBD was improved but not to the degree to bind SARS-CoV-2 RBD. S103F(H)–S33R(L) bound strongly to SARS-CoV-2 RBD with an association rate (ka) of 2.28 × 10^6^ and dissociation rate (kd) of 2.28 × 10^−3^, and to S103Y(H)–S33R(L) with an association rate (ka) of 2.15 × 10^6^ and dissociation rate (kd) of 2.47 × 10^−3^, but CR3022 exhibited a low affinity for SARS-CoV-2 RBD with an association rate (ka) of 1.17 × 10^6^ and dissociation rate (kd) of 1.74 × 10^−2^ (Figure 4E–F).

For SARS-CoV-2 RBD, the SPR results indicated that the affinity of S103F(H)–S33R(L) and S103Y(H)–S33R(L) were ascertained to be enhanced compared to the parent antibody (15-fold and 13-fold, respectively) as shown in Table 2.

### 2.4. Auto-Reactivity of the Wild-Type and Variants of CR3022

The immunofluorescence antibody technique (FAT) was used to determine whether the monoclonal antibody was auto-immune to human Hep-2 cells [25]. It was found that the positive control antibody LC0042 as a known cytoplasmic fibrillar positive control monoclonal antibody showed obvious green specific fluorescence. However, all wild-type and variants of CR3022 showed no immuno-fluorescence response to Hep2 cells even at concentrations up to 100 μg/mL. Moreover, for negative control antibody TNM002 as a negative control of unrelated antigen monoclonal antibody, no green fluorescence was seen (Figure 5). The results showed that wild-type and variants of CR3022 had no immune response to the human Hep-2 cells, and the mutation operation would not affect the safety of the CR3022 antibody.

### 2.5. Structural Stability of Two Double-Site Mutant Antibodies

To explore the molecular mechanism for the mutated antibodies with the enhanced affinity as demonstrated in the above experiments, independent 100 ns MD simulations were performed to further analyze the detailed molecular mechanism of interaction of the CR3022(Fab) and its two double-site mutated antibodies (CR3022–S103F–S33R–RBD and CR3022–S103Y–S33R–RBD) to the SARS-CoV-2 RBD and SARS-CoV RBD at the atomic level.

The root mean square deviations (RMSDs) and the root mean squared fluctuation (RMSF) were subsequently monitored to evaluate the system equilibrium and the structural flexibility. As shown in Figure 6, these two double-site mutated complexes could reach equilibrium rapidly within the initial 10 ns and slightly fluctuate around the 0.28 and 0.22 nm, suggesting that the 100 ns MD simulation could be reasonably covered for simulated systems (Figure 6A). Nevertheless, the RMSD curve of CR3022–SARS-CoV-2 RBD fluctuated more than the other two mutated systems at the first 40 ns and remained stable at 0.18 nm, indicating the structural instability of wild-type CR3022 could be rapidly equilibrated within our MD simulation. The RMSF values of several CDR regions (CDR1, CDR2 and CDR3) of wild-type CR3022 antibody and its two modified antibodies were less than 0.2 nm, showing that these residues involved in the binding interactions were relatively stable with small fluctuations during the entire MD run, which is conducive to tightly binding to the epitope (Figure 6B).

### 2.6. Binding Free Energy Calculation of Wild-Type CR3022 and Two Mutated Antibodies

The equilibrated MD trajectory was selected to calculate the relative binding free energy by using the MM-PBSA method, for analyzing the thermodynamic changes between wild-type CR3022 and its two double-site mutated antibodies. As listed in Table 3, the binding free energy ($\Delta G_{bind}$) of SARS-CoV-2 RBD with the wild-type CR3022 antibody, S103F–S33R and S103Y–S33R complexes were calculated to be −290.01 ± 20.57, −321.63 ± 23.97 and −348.21 ± 17.85 kJ/mol, respectively. Their binding free energy is primarily composed of electrostatic energy, van der Waal energy, and polar/non-polar solvation energy. The calculation results showed that van der Waal energy contributed the most to the binding affinity, followed by electrostatic energy, whereas the polar solvation energy was detrimental to the binding of antigens and antibodies. The MM/PBSA calculation results indicated that the $\Delta G_{bind}$ value of the antibody could be decreased after modification, implying that the binding affinity of double-site mutated antibodies was much stronger than that of the wild-type CR3022.

### 2.7. Per-Residue Energy Decomposition

The per-residue free energy contribution changes to the binding affinity were further analyzed through the per-residue energy decomposition of different CDRs (Figure 7). HCDR–3 was identified to be a crucial region during the binding process. Substituting several residues in this HCDR–3 binding site could give rise to a significant increase in their binding free energy contributions. As shown in Figure 7, a significant increase of 1.8 times was displayed in the 103S mutated to 103F on the HCDR–3 of S103F–S33R. However, the contribution value of 33S mutated to 33F on LCDR–1 could not be significantly increased. It was worth noting that there was a 2.9 times increase in the contribution value of the N35 residue (LCDR–1). We inferred that it might be caused by affecting the structure of its neighboring residue S103F through structural identification. Compared with the S103F–S33R system, the per-residue binding free energy contribution value of the S103Y–S33R system was almost not changed, or even decreases in some sites, and the effect on affinity enhancement might not be as good as the S103F changes.

### 2.8. Distance Analysis of Key Secondary Structures

The previous binding thermodynamic analysis demonstrated that the modified antibodies were more accessible to binding SARS-CoV-2 RBD. It was mainly attributed to the increased RBD accessible surface and more matched structural complementarity. It was found that the distance between two double-site CR3022 mutants and the key secondary structures of SARS-CoV-2 RBD also changed. As displayed in Figure 8A the distance between the secondary structures of the β-sheet2, turn3, and α-helix2 (in the S103F–S33R-RBD and S103Y–S33R–RBD systems) and the CDR region of the CR3022 was significantly reduced compared with the wild-type CR3022–SARS-CoV-2 RBD system. The distance fluctuations of wild-type CR3022 could be around 1.1–1.2 nm, whereas that of two modified antibodies fluctuated were around 1.0–1.05 nm. It was obvious that the distance of these key residues involved in the binding interactions between CDR of the modified antibody and key secondary structures was much closer than that of the wild-type CR3022 during the MD run (Figure 8B). These calculation results were consistent with the experimental observation of affinity enhancement. Nevertheless, as the ribbon diagram of SARS-CoV-2 RBD–S103Y–S33R was also shown in Figure 8, the rearrangement that occurred in the epitope was not particularly significant, except for α-helix2.

### 2.9. The Binding Profiles Analysis

The non-bonded interactions were essential for antibody-antigen complex overall structural stability and binding dynamics. Based on the conformational analysis, the residue of 33S mutated to 33R on CR3022 was beneficial to the HCDR–3 structural optimization in the S103F–S33R system (Figure 9B,C). The original serine was relatively small and away from its epitope on the binding interface, whereas the substituted arginine exhibited a larger guanidyl sidechain than that of serine. The R33 extended sidechain with perfect structural flexibility and hydrophobicity was supposed to be essential for the hydrogen-bonded formation. Its nitrogen atom with the oxygen atom of D428 in the RBD forms an ionic bond (R33:NH1–D428:OD2) to improve the binding interaction between the antibody and antigen. A new hydrogen bond was also formed between the S33R and T430 (R33:HD2–T430:OG1) (Figure 9B). As shown in Figure 9, a π–π hydrophobic interaction was also formed between the two benzene rings of F103 in the heavy chain and Y31 in the light chain. This promoted the stronger assembly between the heavy and light chains of the antibody. In addition, with the displacement of Thrn3, the K386 residue of RBD formed a new salt bridge with the oxygen atom of E61 in the light chain(K386:HZ1–E61:OE2) and K386 maintained a salt–bridge with the Y55 residue(K386:HA–Y804:OH). Meantime, the hydrophobic interaction was formed between A370 and Q1 (Q1:H2–A370:OD1). The synergistic effects of these non-bonded interactions formed on the binding interface surface made its binding affinity significantly stronger.

In the S103Y–S33R system, the S33 residue was replaced by R33. R33 acted as a bridge between the S103Y–S33R and SARS-CoV-2 RBD. The R33 residue and the E516 residue of the RBD were close to each other and exhibited opposite charges, which could provide a favorable environment for electrostatic interaction(R33:NH1–E516:OE2). Similarly, it also formed an electrostatic interaction with the D428 residues in such a favorable environment(R33:NH2–D428:OD1). However, the hydrogen-bonded interaction between the S33 and T430 residue was found to be lost in the wild-type CR3022 antibody system. The hydrophobic interaction between the A372 and Y27 residues could remain and an additional hydrogen bond was formed between both of them(A372:H–Y27:OH). These key binding sites were located on the interface of SARS-CoV-2 RBD and the CDRs of CR3022.

## 3. Materials and Methods

### 3.1. Structure Preparation

The structure of the CR3022(Fab) SARS-CoV-2 RBD complex resolved by the X-ray experiment has been utilized as the initial structure of this complex. This complexed structure downloaded from the RCSB PDB database (PDB: 6W41) is crystallized at a moment in the binding dynamic process [15]. This static structural information may not provide the real structure that actually interacts with each other. In order to obtain the global optimized structure of the complex, we first performed a 100 ns MD simulation on the initial structure using the Discovery Studio (DS) platform [26]. From the balanced MD trajectory, its representative structure “N76200” as a global optimized structure was extracted. The detailed structural optimization steps were inconsistent with our previous simulation [16]. The CHARMM27 force field [27] was used in the Generalized Born with a simple Switching (GBSW) solution model [28]. The energy minimization (EM) was performed by 1000-step the Steepest Descent (SD) and Conjugate Gradient (CG) methods on the DS module [29,30] in Discovery Studio 4.5 software package [23,24]. The non-bonded interaction value was set to 1.4 nm [31]. The output structures with a convergence value lower than 0.4184 kJ/mol·nm were taken for the initial conformation of the next MD study.

### 3.2. Virtual Mutation

Site-directed mutagenesis is an important tool to investigate the relationship between protein structure and function [24,32]. With the help of the computer-guided virtual mutation (VM), the amino acid sites in the antibody-CDRs were mutated based on the conformational and energy changes after the mutation, and each amino acid in the CDR region was radically mutated to other 19 amino acids. Virtual mutations have been performed on the antibodies from the crystal structure 6W41 and from the representative structure N76200. The amino acid in the CDR region of CR3022 was mutated using the calculated Binding-Mutation-Energy module of the Discovery Studio 4.5 software package [23,24]. The changes of binding energy were calculated to screen for increased antigen-antibody binding. If the calculated Mutation Energy (Δ*G*) < −1 kJ/mol after mutation, it means the binding of the mutated structure increased [22]. Compared to the original amino acid, the replacement of new amino acids reduced the binding energy between antigen and antibody and was conducive to the binding of antigen and antibody. Conversely, if the calculated Mutation Energy (Δ*G*) > 1 kJ/mol after mutation, it indicated that the new residue may play an important role, which this hotspot is the key residue on the antigen-antibody binding interface. For the purpose of this study, we selected the sites with calculated Mutation Energy (Δ*G*) < −1 kJ/mol as our candidate site for further verification. Finally, the Build Mutants module of Discovery Studio 4.5 software package [24] was used to construct single and double-site mutants systems.

### 3.3. Protein Expression and Purification

Further experimental verification was carried out by selecting sites that were more favorable for CR3022 binding to SARS-CoV RBD after mutation. The gene of antibodies and antigens were synthesized by BGI. The specific primers containing mutation sites were designed by NEB base changer (https://nebasechanger.neb.com/, accessed on: 19 May 2021), and the mutation was introduced by the Q5 site-directed mutation kit (Cat# E0552S). The SARS-CoV and SARS-CoV-2 RBD amino acid sequences were obtained from GeneBank with its corresponding ID numbers: NP_828851.1 (SARS-CoV-2 RBD) and YP_009724390 (SARS-CoV RBD). Eukaryotic cells 293i (Cat# A14527) were employed to express proteins efficiently, and the supernatant purified samples were collected after 5 days. Antigen and antibody were purified with the purity of 99% or above by Ni-NTA agarose (Cat# 17524801) and Protein A Resin (Cat# L00210), respectively. Enzyme-linked immunosorbent assay (ELISA) and surface plasmon resonance (SPR) were used to detect the binding ability of antibodies and antigens.

### 3.4. Antibody Binding Detection by ELISA

Microtiter plates were coated with SARS-CoV RBD or SARS-CoV-2 RBD at a concentration of 1 μg/mL in coating solution (100 μL/well). The coated plates were sealed and incubated overnight at 4 °C, and then washed with 0.1% PBS/Tween-20 (PBST) solution three times and sealed with 300 μL/well of blocking solution (PBS containing 5% goat serum). They were incubated at room temperature for 120 min. Next, 100 μL of wild type or mutant CR3022 antibodies at 4 μg/mL were added to the first wells followed by three-fold dilutions and incubated for 60 min. Then, the horseradish peroxidase-labeled sheep anti-human IgG (IgG1–HRP) was added as a secondary antibody and incubated for 30 min. Plates were washed five times with PBST. Finally, 100 μL of TMB (Invitrogen, Thermo Fisher Scientific, Waltham, MA, USA) color developer was added, and 5 min later the reaction was stopped by 1.5 M H_2_SO_4_. The OD values were measured at 450 nm. Triplicate experiments were performed for each sample.

### 3.5. Antibody Affinity Detection by SPR

SPR analyses were performed on a Biacore 8k instrument (GE Healthcare Bio-Sciences AB). CM5 sensor chip (Cat# BR100012) was conjugated with purified anti-human antibody by using an amine coupling procedure. HBS-EP (10 mM HEPES, pH 7.4, 150 mM NaCl, 3 mM EDTA, 0.005% surfactant P20) was used as reaction buffer at a flow rate of 10 µL/min. The affinity constant (K_D_) is equal to the ratio of the rate constant (K_D_ = kd/ka, the association rate constant (ka), dissociation rate constant (K_d_)), antigen-antibody with 1:1 kinetics model binding. CR3022 and a series of mutated antibodies were captured by anti-human antibody immobilized on CM5 chips for measurement of the binding to SARS-CoV RBD and SARS-CoV-2 RBD proteins. Five 2-fold serial dilutions of SARS-CoV RBD and SARS-CoV-2 RBD proteins starting at 50 nM or 100 nM in HBS-EP were injected at a rate of 30 mL/min for 90 s with a 300 s dissociation. The surface of the chip was regenerated with 5 nM MgCl2. All mathematical and fitting operations were performed using the BIA4.1 assessment software.

### 3.6. Auto-Immune Reactivity Test

The immunofluorescence antibody technique (FAT) was used to detect whether the wild-type and variants of CR3022 had autoimmune reactivity. First, the Hep-2 cells grown to 80–90% in a 96-well plate (Costar, Cat# 3599) were gently washed twice with PBST and air-dried, then 100 μL of 80% acetone was added and fixed at room temperature for 15min. After washing with PBST, 300 μL of 5% goat serum blocking solution were added followed by incubation in a 37 °C incubator for 1 h. Then 50 μL of antibodies (CR3022, S103F–S33R and S103Y–S33R), negative control antibody TNM002, positive control antibody LC0042 at a concentration of 100 μg/mL, 50 μg/mL and 25 μg/mL were added to cells in the microtiter plates and incubated for 1 h at 37 °C. Finally, after washing three times with PBST, a 50 μL secondary antibody (Solebo Cat# SF101) of goat anti-human IgG-FITC diluted at 1:500 was added. After incubating at 37 °C for 50 min, they were washed five times with PBST, observed and photographed under EVOS fluorescence microscope (Thermo).

### 3.7. Molecular Dynamics (MD) Simulation

Based on the Build Mutants module in the DS4.5 software package [26], we constructed the complex structure of the double-site mutants for the following MD calculation and binding thermodynamic and dynamic analysis. By MD simulation, the conformation of the CR3022–SARS-CoV RBD system could be globally optimized, and the MD trajectories can also be used to calculate binding free energy and dynamic process. Here the open source software, Gromacs5.1.2 package [33,34] was utilized for the system construction and MD calculations. The optimized structure was fully dissolved in a rectangular water box filled with TIP3P (size 10.0 Å × 14.5 Å × 10.0 Å) [35,36]. The RSFF2C force field was applied for MD calculation [37]. All systems were equilibrated by NVT and NPT for 500 ps [29,38]. The 100 ns MD simulations were performed at the periodic boundary conditions of 310 K and 1 bar [39]. To keep all bonds rigid, the bond lengths of protein and water molecules were constrained with LINCS and SETTLE algorithm [40], respectively, the electrostatic interaction was calculated by Particle Mesh Ewald (PME) method [41], and Verlet [42] was used for truncation. The above operations were performed by the Gromacs 5.1.2 software package. Root-mean-squared deviation (RMSD), root-mean-squared fluctuation (RMSF), groups distance and non-bonded interactions, such as hydrogen-bonding and hydrophobic interactions as well as salt-bridge were analyzed during MD trajectories.

### 3.8. Binding Free Energy Calculation by MM/PBSA

Molecular mechanics/Poisson–Boltzmann surface area (MM-PBSA) is a common method to calculate the binding free energy of ligand and receptor [43,44,45,46,47,48]. Using g_mmpbsa software [45], 100 conformations were extracted from the MD equilibrium trajectories for calculating the binding free energy ($\Delta G_{bind}$). The interaction energies, like the free solvation and intra-molecular energy, involved in binding were evaluated by using the following equation:Δ*G**_bind_* = Δ*E**_MM_* + ΔΔ*G**_sol_* − *T*Δ*S*
= Δ*E**_MM_* + Δ*G**_PB_* + Δ*G**_SA_* − *T*Δ*S*
where Δ*G_bind_* is the binding free energy, Δ*E_MM_* is the potential energy of electrostatic and van der Waals difference in a vacuum; ΔΔ*G_sol_* is the solvation free energy difference, including polar solvation energy (Δ*G_polar_*) and non-polar solvation energy (∆*G_np_*). ∆*G_polar_* can be obtained by solving the linear Poisson–Boltzmann equation (PB), and ∆*G_np_* can be determined by calculating the molecular surface area (SA); T is the absolute temperature and ∆S is the entropy change [49]. The entropic contribution (*T*Δ*S*) is expected to be similar for the three systems and can be neglected [48,50].

## 4. Conclusions

Site-directed mutagenesis is an important method to study the relationship between protein structure and function. Computer-simulated mutation can be used to quickly and accurately obtain favorable mutation sites and realize high-throughput virtual screening. During the process of affinity maturation of natural antibodies, the high-frequency mutations of somatic cells are mainly focused on the antibody CDR regions that directly interact with the antigen. Here we provided an effective strategy to improve the affinity of the CR3022 antibody to SARS-CoV-2 RBD.

With the aid of computer simulation calculation, the CDR regions of the CR3022 antibody were subjected to saturation mutation, and the identified favorable mutation points were selected based on the mutation energy (Δ*G_bind_*) value after the mutation for the next step of experimental confirmation and analysis. Ten sites on the heavy chain and eight on the light chain meet screening conditions.

ELISA and SPR experiments showed that S103F and S103Y on the heavy chain and S36R on the light chain can improve the affinity of CR3022, but improvement is limited. Thus, we produced additional antibodies with double-site mutations named S103F–S33R and S103Y–S33R. Compared with the wild-type affinity of CR3022, S103F–S33R and S103Y–S33R with double-site mutations bound to SARS-CoV-2 RBD with enhanced affinity by 15-fold and 13-fold, respectively. The autoreactivity test confirmed that variants of CR3022 did not bring autoimmune reactivity in Hep2 cell staining assays.

The MD trajectory analysis indicated that the distance between three secondary structures (α-helix2, β-sheet2 and β-turn3) on the SARS-CoV-2 RBD and the CDR regions of these two double-site mutants were shortened. The proximity of these key sites in the binding interface resulted in the participation and formation of additional non-covalent bonded interactions, such as new hydrogen bonds, salt bridges and hydrophobic interactions. Consistent with previous experimental results, the binding free energy calculation and per-residue energy decomposition results also showed that these two favorable double-site mutants could bind more strongly to SARS-CoV-2 RBD than that of wild-type CR3022.

The above results clearly showed that the affinity of the CR3022 antibody modified based on computer simulation guidance has been greatly enhanced which suggest a potential direction for the development of high-affinity antibodies against coronaviruses. At the same time, we have also noticed that although the affinity of variants of CR3022 was markedly enhanced, the neutralizing activity in vitro was not much improved (Data not shown). The affinity of an antibody is not directly equivalent to in vitro neutralizing ability. Likewise, neutralizing capacity in vitro is not directly equivalent to protective capacity in vivo. Therefore, the improvement of the neutralizing activity of the CR3022 antibody still needs to be further explored.

Overall, this study applied computational and experimental approaches to study the affinity maturation of CR3022 antibody in vitro, the findings provide useful insights for the development of anti- SARS-CoV-2 antibody drugs for the treatment of COVID-19.

## Figures and Tables

**Figure 1 viruses-14-00186-f001:**
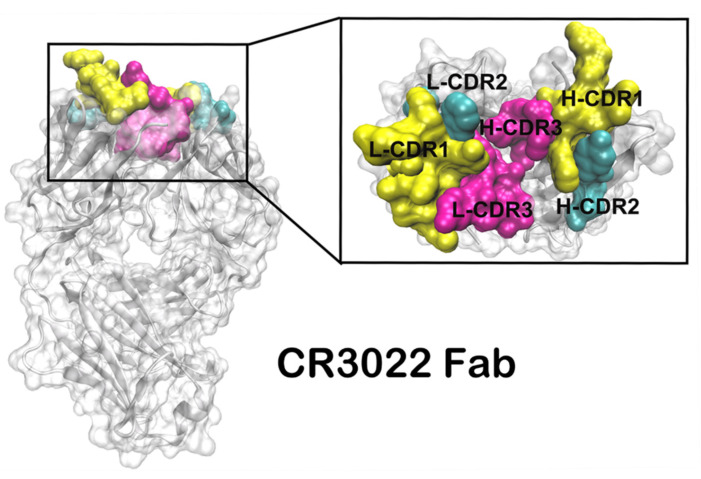
The ribbon diagram of the binding mode of CR3022.The light chain and heavy chain CDR1(in yellow), CDR2(in cyan), CDR3(in magenta) were highlighted in the cartoon. The image of the 3D spatial structure was generated using the Discovery Studio 4.5 software package.

**Figure 2 viruses-14-00186-f002:**
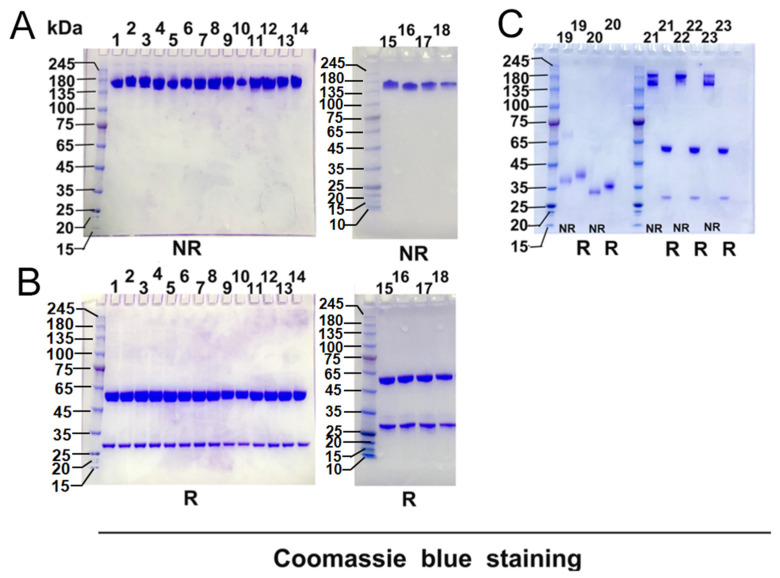
Recombinant CR3022 antibody, CR3022 variants and RBD analyzed by SDS-PAGE. All proteins were fractionated 4–12% Bis-Tris SDS-PAGE under both non-reduced and reduced conditions. Gels were stained with Coomassie blue dye. All antibodies and RBD protein migrated in the gels as a single protein band with an apparent MW of 160 kDa(antibody) or 30–38 kDa(RBD) under non-reduced condition (NR) (**A**) and as 2 protein bands with apparent MWs of 50 kDa and 30 kDa under reduced conditions (R) (**B**,**C**). The sample information of (**A**) 1–18 lanes were non-reduced of G28W, T31F, T31Y, S103R, S103I, S103F, S103W, S103Y, S33R, S33H, S33I, S33L, S33K, S33M, S33F, S33W, S33Y, N35Y. The sample information of (**B**) 1–18 lanes were reduced of G28W, T31F, T31Y, S103R, S103I, S103F, S103W, S103Y, S33R, S33H, S33I, S33L, S33K, S33M, S33F, S33W, S33Y, N35Y. The sample information of (**C**) 19–23 lanes were non-reduced and reduced of SARS-CoV RBD, SARS-CoV-2 RBD, CR3022, S103F(H)–S33R(L) and S103Y(H)–S33R(L), respectively.

**Figure 3 viruses-14-00186-f003:**
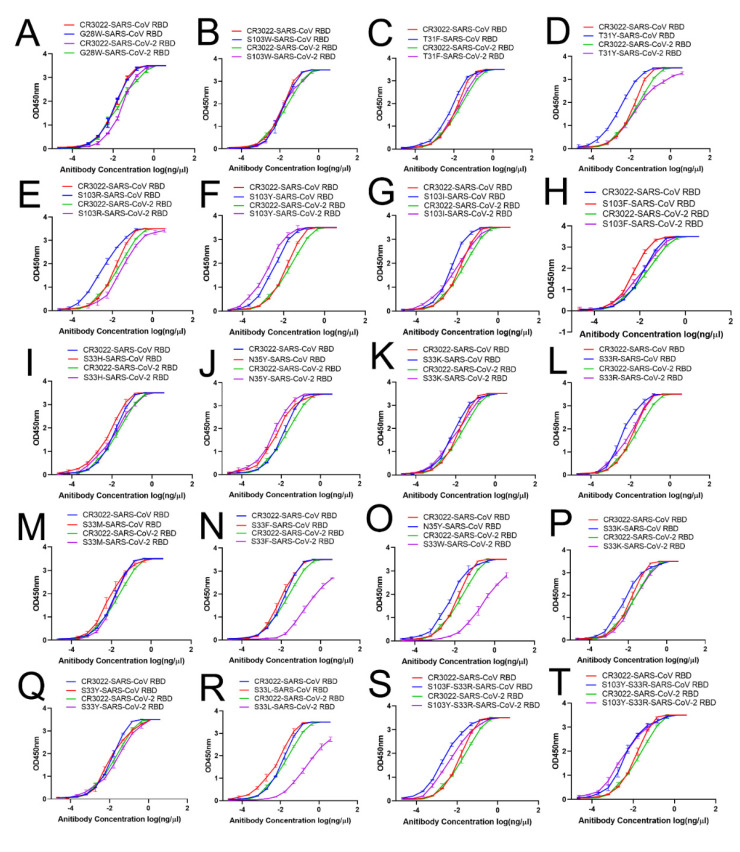
Analysis of the ability of 18 single-site mutants and two double-site mutated antibodies to bind SARS-CoV RBD and SARS-CoV-2 RBD with wild-type in ELISA. The ability of the wild-type and mutant antibodies as indicated in the individual binding curves were tested in 12-fold serial dilutions for binding to SARS-CoV RBD and SARS-CoV-2 RBD. The binding curves of (**A**–**T**) were single-site mutant of G28W, T31F, T31Y, S103R, S103I, S103F, S103W, S103Y, S33R, S33H, S33I, S33L, S33K, S33M, S33F, S33W, S33Y, N35Y and two double-site mutants of S103F(H)–S33R(L) and S103Y(H)–S33R(L), respectively.

**Figure 4 viruses-14-00186-f004:**
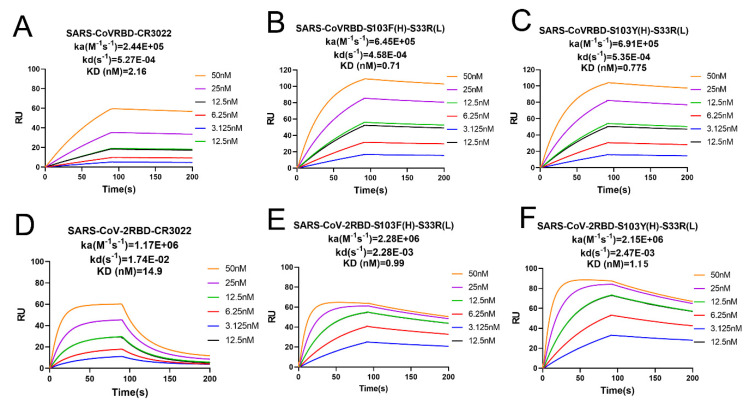
The binding kinetics and rate constants of SARS-CoV RBD and SARS-CoV-2 RBD to the wild-type CR3022 and its double-site mutation antibodies measured by surface plasmon resonance (SPR). SARS-CoV RBD-CR3022 (**A**), SARS-CoV RBD-S103F(H)–S33R(L) (**B**), SARS-CoV RBD-S103Y(H)–S33R(L) (**C**) and SARS-CoV-2 RBD-CR3022 (**D**), SARS-CoV-2 RBD-S103F(H)–S33R(L) (**E**) and SARS-CoV-2 RBD-S103Y(H)–S33R(L) (**F**) are the binding curves with decreasing concentrations of the indicated antibodies injected over the antibody captured by anti-Fc tag antibody immobilized on an SPR chip. Values of kinetic rate constants (KD(nM)) were indicated for each antibody on top of the individual plots.

**Figure 5 viruses-14-00186-f005:**
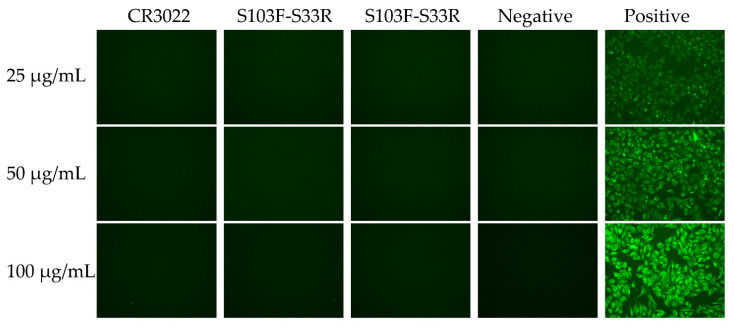
Autoimmune reactivity of wild-type and variants of CR3022 (CR3022, S103F–S33R and S103Y–S33R) to human Hep2 cells with TNM002 as negative control and LC0042 as a positive control. Green fluorescence was photographed under an EVOS fluorescence microscope. The three antibodies to be tested (CR3022, S103F–S33R and S103Y–S33R) and negative control antibody TNM002, positive control antibody LC0042 were added at three concentrations of 25 μg/mL, 50 μg/mL and 100 μg/mL, respectively.

**Figure 6 viruses-14-00186-f006:**
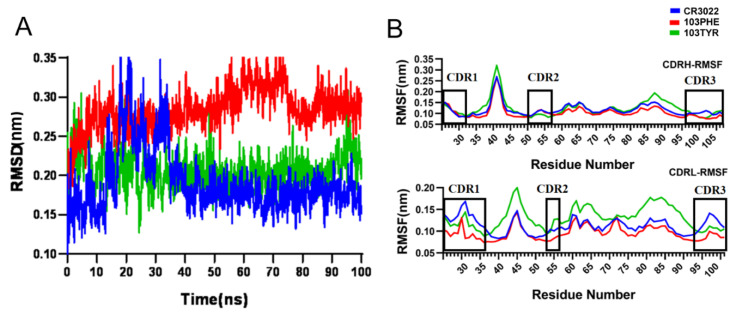
The time-dependent RMSD (**A**) and RMSF (**B**) curves of wild-type CR3022–SARS-CoV-2 RBD system (in blue) and its two double-site mutated complexes (S103F–S33R–RBD, in red and S103Y–S33R–RBD, in green) during 100 ns MD simulation. The approximate CDR locations in the heavy and light chain (CDR1, CDR2 and CDR3) with IMGT number were respectively marked with a black frame.

**Figure 7 viruses-14-00186-f007:**
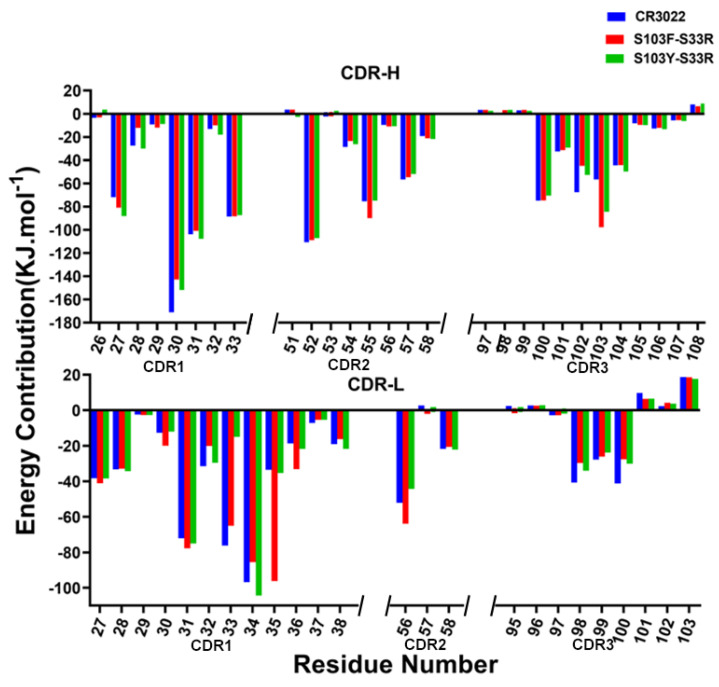
The per-residue binding free energy contributions of different CDRs in the heavy (above) and light (below) chains. The sequence of CDR1, CDR2 and CDR3 with IMGT numbers were separated by a break.

**Figure 8 viruses-14-00186-f008:**
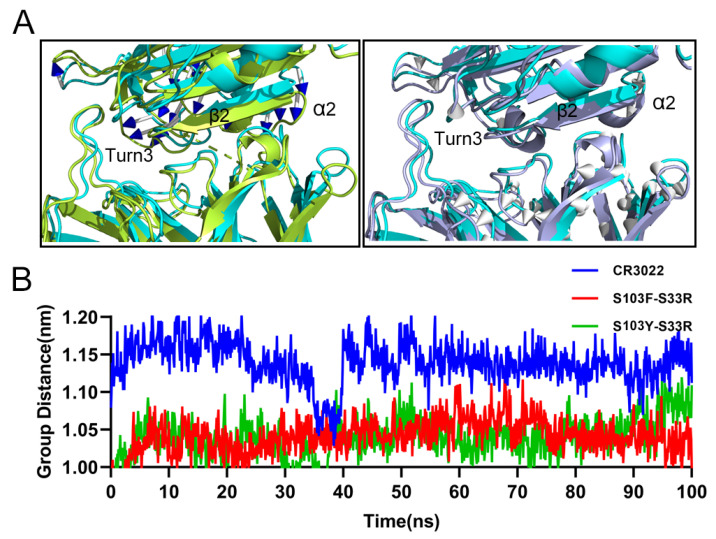
The time-dependent distance changes analysis between the CDR group and the epitope group. The modevectors.py script was used to visualize the direction of epitope motion from the starting structure to the final structure. SARS-CoV-2 RBD(in cyan) was specified as a starting structure, S103F–S33R–RBD (in yellow) and S103Y–S33R–RBD (in purple) were specified as a final structure. The arrows were connected between the initial and final states, indicating the direction of epitope motion (**A**). In addition, the change of distances between the CDR and epitope was also calculated during the 100 ns MD simulation (**B**).

**Figure 9 viruses-14-00186-f009:**
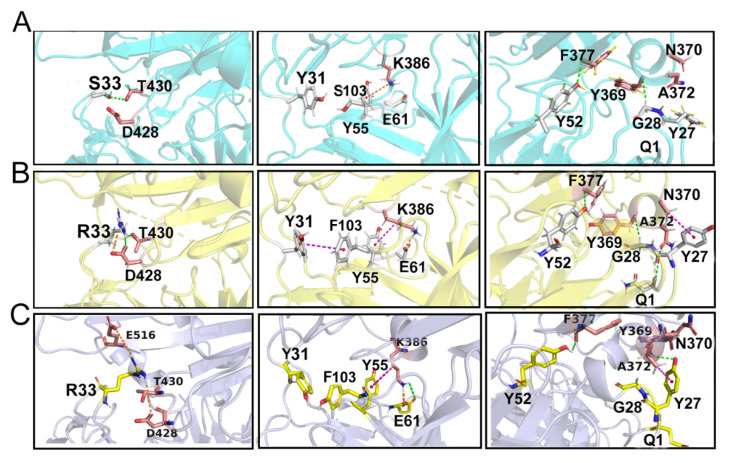
Hydrogen-bonded and hydrophobic interaction network analysis. Three secondary structures (α-helix2, β-sheet2 and β-turn3) on the SARS-CoV-2 RBD participated in non-covalent bonded interaction with the CDR regions of the antibodies. This specific hydrogen, electrostatic and hydrophobic interaction of CR3022 SARS-CoV-2 RBD (**A**, in cyan), S103F–S33R–RBD (**B**, in yellow) and S103Y–S33R–RBD (**C**, in purple) with IMGT number were represented by the green dashed line, gold dashed line and magenta dashed line, respectively.

**Table 1 viruses-14-00186-t001:** The favorable mutation sites were filtered by the Δ*G_bind_* value of < −1 kJ/mol after saturation mutagenesis.

Location	Mutation Sites (IMGT Number Scheme)	Δ*G_bind_* (kJ/mol)	
		6W41	N76200
Heavy Chain CDR1	G29W	−1.54	−1.29
	T36F	−1.11	−2.48
	T36Y	−1.07	−1.14
Heavy Chain CDR3	S103R	−2.21	−2.96
	S103I	−1.84	−1.25
	S103F	−2.05	−2.04
	S103W	−1.35	−1.87
	S103Y	−3.13	−2.43
Light Chain CDR1	S33R	−2.62	−2.94
	S33H	−1.34	−2.41
	S33I	−1.69	−2.13
	S33L	−2.31	−2.75
	S33K	−1.96	−1.67
	S33M	−1.47	−1.47
	S33F	−1.81	−2.92
	S33W	−2.38	−2.95
	S33Y	−2.35	−2.29
	N35Y	−2.01	−1.48

**Table 2 viruses-14-00186-t002:** The binding affinity (KD), association rate (ka) and dissociation rate (kd) value of CR3022 and its double-site mutation antibodies to SARS-CoV RBD and SARS-CoV-2 RBD.

Analyte Solution	Capture Solution	Ka (M^−1^s^−1^)	Kd (s^−1^)	KD (M)
SARS-CoV RBD	CR3022	2.44 × 10^5^	5.27 × 10^−^^4^	2.16 × 10^−^^9^
	S103F(H)–S33R(L)	6.45 × 10^5^	4.58 × 10^−^^4^	7.10 × 10^−^^10^
	S103Y(H)–S33R(L)	6.91 × 10^5^	5.35 × 10^−^^4^	7.75 × 10^−^^10^
SARS-CoV–2 RBD	CR3022	1.17 × 10^6^	1.74 × 10^−^^2^	1.49 × 10^−^^8^
	S103F(H)–S33R(L)	2.28 × 10^6^	2.28 × 10^−^^3^	9.99 × 10^−^^10^
	S103Y(H)–S33R(L)	2.15 × 10^6^	2.47 × 10^−^^3^	1.15 × 10^−^^9^

**Table 3 viruses-14-00186-t003:** The binding free energy of the SARS-CoV-2 RBD with the wild-type CR3022 antibody, S103F–S33R and S103Y–S33R complexes.

Complex	Van der Waal Energy (Δ*G_vdw_*)	Electrostatic Energy (Δ*G_ele_*)	Polar Solvation Energy (Δ*G_PB_*)	SASA Energy (Δ*G_SA_*)	Binding Energy (Δ*G_bind_*)
SARS-CoV-2 RBD–CR3022	−401.25 ± 2.59	−261.41 ± 3.35	778.63 ± 12.93	−46.18 ± 0.38	−290.01 ± 20.57
SARS-CoV-2 RBD–S103F–S33R	−451.37 ± 2.62c	−328.87 ± 4.42	905.99 ± 13.23	−50.98 ± 0.37	−321.63 ± 23.97
SARS-CoV-2 RBD–S103Y–S33R	−445.01 ± 2.88	−303.79 ± 2.89	825.26 ± 8.59	−51.87 ± 0.38	−348.21 ± 17.85

All values are given in kJ/mol, with corresponding standard errors of mean in “±”.

## Data Availability

All data generated or analyzed during this study are included in this published article.

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
