# Peer review of "Structure Based Affinity Maturation and Characterizing of SARS-CoV Antibody CR3022 against SARS-CoV-2 by Computational and Experimental Approaches"

_viruses, 2022, doi:10.3390/v14020186_

Round 1
Reviewer 1 Report
In their article „Structure Based Affinity Maturation and Characterizing of SARS-CoV antibody CR3022 against SARS-CoV-2 by Computational and Experimental Approaches“ Yu et al. use computational approaches to identify mutations within the SARS-CoV-neutralizing antibody CR3022 that could improve its SARS-CoV-2 cross-reactivity. They identified 10 single heavy chain and 8 single light chain mutations in-silico and expressed those recombinantly together with two double-site mutations and the wild-type CR3022 antibody. Recombinant CR3022 variants were experimentally tested for avidity by ELISA and affinity by SPR and confirmed some of their in-silico findings in terms of improved affinity. Finally, they elucidate the underlying molecular mechanism by molecular dynamics.
This reviewer absolutely agrees that cross-neutralizing antibodies will likely be a major component in the fight against the the ongoing SARS-CoV-2 pandemic. As a major strength, this manuscript provides key mutations in CR3022, which increase its affinity to SARS-CoV-2 and could thus improve its applicability as a therapeutic antibody against Sarbecoviruses. However, while an interesting approach in principle, this reviewer has major concerns and comments, which are listed below.
General comments:
The manuscript needs major editorial revisions in terms language. Several flaws in grammar/vocabulary as well as numerous typos make it extremely hard to understand and grasp the rational behind experimental design as well as the authors conclusions. Similarly, several figures are erroneous and should be revised to make it easier to understand the data (also see specific comments). In addition, references should be checked again by the authors as many of the bioRxiv papers could be substituted by recent articles in high-ranked peer-reviewed journals and certain references in the introduction seem inappropriate. In this respect, the authors should also cite the original article, where CR3022 has been isolated.
With respect to the overall structure of the manuscript, it is not clear in the first place, why the double mutants were chosen. It seems that the rational comes from the SPR experiments. However, the double mutants are introduced in the very first results and at this point this reviewer was struggling to understand the selection process of the double mutants. The authors should elaborate on the rational behind the double-mutant selection.
In addition numbering of amino acid positions switch between IMGT and PDB-file numbering, which makes it hard to follow their argumentation and keep track of identifying critical residues. The authors should stick to IMGT numbering and/or provide PDB as additional information.
There are two major scientific issues, which should be addressed. First, artificial mutations might lead to auto-reactivity in vivo, which would prevent the therapeutic application of CR3022 against Sarbecoviruses as suggested by the authors. The authors should perform experiments to investigate if mutant CR3022 variants are prone to auto-reactivity. Second, it is well documented that affinity is not correlated with neutralization in the way that there are plenty of highly-affine antibodies, which are non-neutralizing. The authors need to elaborate on their conclusion why the introduction of increased affinity will lead to improved SARS-CoV-2 neutralizing antibodies. In the current version of the manuscript, they only tested for avidity and affinity but not for neutralization.
Specific comments:
Line 37: Needs to be clarified since no experimental evidence is given in the manuscript for improved neutralization (see general comments).
Line 110: Which physicochemical properties have been tested? How do the authors define „improved physicochemical properties“?
Lines 117-122: It is not clear here, why the double mutant was chosen (see general comments).
Figure 2: For the gel on the right it is not clear, which lanes were reduced and non-reduced. In addition, the figure Seems to be a composite of several gels. There should be a ladder added to independent gels. It is not clear, why the signals in lanes 15-18 in the second gel (R) are higher than in lanes 1-14.
Line 139: Lanes 1-18 are mentioned, but lanes 1-23 are shown in the figure.
Line 146: „could not affect its binding“. This reviewer doubts this conclusion. Looking at the figures, I see up to 1 log-fold change in EC50.
Figure 3 A/B/C/D: OD450 should probably read nm (nanometers) not nano molar (nM)
Figure 3 A/B/C/D: antibody concentration in ng/ul. Greek micro needs to be replaced here.
Figure 3 A/B/C/D: Scales are probably log scales. Either add power of 10 no the axis or log in the axis label.
Line 153: „muTable 2.3“ The figure legend does not make sense.
Line 158: Reference to Figure and previous reports is missing.
Line 177-178: This sentence needs revision. Not clear, what was compared to what here.
Line 186: How was significance tested? If no test was performed, „significantly“ should be replaced.
Figure 4: KD in A should be given as nM as all other KDs in this figure.
Figure 4 D/E: kd is equal between D and E and does not correspond between table 2 and 4E.
Figure 5B: IMGT Numbering should be given (either in addition or as replacement for PDB numbering)
Line 243: „N78“ not visible from Figure 6 since different numbering was used.
Figure 6: should include IMGT numbering and label CDR1, 2 and 3
Lines 356-358: Should add the reference for SARS-CoV and SARS-CoV-2 RBD sequences and exactly define which amino acids were cloned for production.
Line 448: How was significance tested?
Line 466: This sentence seems to make no sense here.
Author Response
Response for Reviewer 1:
General comments
General comments 1: The manuscript needs major editorial revisions in terms language. Several flaws in grammar/vocabulary as well as numerous typos make it extremely hard to understand and grasp the rational behind experimental design as well as the authors conclusions.
Response: Thanks for the suggestion, and we have corrected the errors throughout the manuscript. We also apologize for the poor language in some sections of the manuscript, especially for the conclusion section, and believe that in some cases it might have misconstrued our intended meaning. Our revised manuscript has been edited and polished, and we believe that in this version, most of the language flaws have been fixed.
General comments 2: Similarly, several figures are erroneous and should be revised to make it easier to understand the data (also see specific comments).
Response: We have carefully checked and corrected all figures and tables, thanks for the suggestion.
General comments 3: In addition, references should be checked again by the authors as many of the bioRxiv papers could be substituted by recent articles in high-ranked peer-reviewed journals and certain references in the introduction seem inappropriate.
Response: Thanks for the suggestion. It has been corrected in this version.
General comments 4: In this respect, the authors should also cite the original article, where CR3022 has been isolated.
Response: It has been corrected in this version. The original article isolating CR3022 has been cited.
General comments 5: With respect to the overall structure of the manuscript, it is not clear in the first place, why the double mutants were chosen. It seems that the rational comes from the SPR experiments. However, the double mutants are introduced in the very first results and at this point this reviewer was struggling to understand the selection process of the double mutants. The authors should elaborate on the rational behind the double-mutant selection.
Response: Thanks for the suggestion. It has been added in the revised manuscript as follow:Page 4, Lines: 124-237: “It was found that when the S103 residue on Heavy Chain CDR3 was mutated to five other amino acids, the stability of antigen-antibody binding could be enhanced. The same was true when the S33 residue on Light Chain CDR1 was mutated to other nine amino acids in this way. In order to explore the theoretical feasibility mentioned in this article, synergistic effect brought by multiple mutations and the internal interaction between the sites need to be considered at the same time. On the other hand, considering the complexity of antigen-antibody interactions, their binding interface might involve various non-bonded interactions formed by mutiple residues. It was unlikely that changing only a single residue in a tremendous protein system will much more improve the binding affinity of antibody. Based on the results obtained by computational virtual results of single-point mutations, the multi-site mutations on the CR3022 antibody were needed to further optimize the antigen-antibody interaction interface.”
General comments 6: In addition numbering of amino acid positions switch between IMGT and PDB-file numbering, which makes it hard to follow their argumentation and keep track of identifying critical residues. The authors should stick to IMGT numbering and/or provide PDB as additional information.
Response: It has been corrected in this version. IMGT numbering of amino acid positions is used.
General comments 7: There are two major scientific issues, which should be addressed. First, artificial mutations might lead to auto-reactivity in vivo, which would prevent the therapeutic application of CR3022 against Sarbecoviruses as suggested by the authors. The authors should perform experiments to investigate if mutant CR3022 variants are prone to auto-reactivity.
Response: The corresponding experiments have been added to this revised manuscript in the revised manuscript as section 2.4 and section 3. The results showed that wild-type and variants of CR3022 had no immune response to the human Hep-2 cells, and the mutation operation would not affect the safety of CR3022 antibody.
General comments 8: Second, it is well documented that affinity is not correlated with neutralization in the way that there are plenty of highly-affine antibodies, which are non-neutralizing. The authors need to elaborate on their conclusion why the introduction of increased affinity will lead to improved SARS-CoV-2 neutralizing antibodies. In the current version of the manuscript, they only tested for avidity and affinity but not for neutralization.
Response: Corresponding neutralization experiments have been added to this revised manuscript, and although the neutralization titer has not been significantly improved, the ability to inhibit the virus has been improved.
At the same time, we also noticed that there are reports that the CR3022 antibody itself does not have neutralizing activity against SARS-CoV-2 in vitro. However, it cannot be deduced from this that it has no protective effect in vivo.
The above conclusion can be confirmed from a published Science article (Yuan, M. et al. Science 368, 630-633) which mentioned in the article: In the in vitro micro-neutralization experiment, the concentration of CR3022 reached 400 μg/mL, and the effect of neutralizing the SARS-CoV-2 virus was not seen, but it cannot be said that CR3022 cannot protect individuals against SARS-CoV-2 in vivo.
Due to the specificity of the CR3022 binding epitope which does not overlap with the RBD-ACE2 binding site, may lead to the generation of a "breathing pattern". This situation is similar to that antibodies targeting the surface epitope of the HA trimer of influenza A virus do not neutralize in vitro, but protect individuals against influenza A virus attack in vivo. (Bangaru, S. et al. Cell 177, 1136-1152 e1118; Watanabe, A. et al. Cell 177, 1124-1135 e1116; Bajic, G. et al. Cell host & microbe 25, 827-835 e826)
Therefore neutralization experiments in vitro cytology are not directly equivalent to the protective effect of antibody drugs in vivo,the purpose of our article is to improve affinity, but affinity is not equal to neutralization ability in vitro, but it does not mean that there is no protective effect in vivo.
In conclusion, the combination of computational and experimental approaches in this article can provide insights for developing as broadly neutralizing antibody drugs against SARS-CoV-2, which is the main purpose and value of the article.
Specific comments:
- Line 37: Needs to be clarified since no experimental evidence is given in the manuscript for improved neutralization (see general comments).
Response: It has been added in this version. Please see the response of “general comments 8”
- Line 110: Which physicochemical properties have been tested? How do the authors define „improved physicochemical properties“?
Response: Thank you for your suggestion.We have revised the corresponding text, “virtual mutation (VM) were performed using the calculated Binding Mutation Energy module of Discovery Studio 4.5 software package,The CR3022 mutants with improved affinities and physicochemical properties were obtained with the mutation energy (ΔGbind) less than -1 kJ/mol as the screening criterion which reflected that the mutant took higher binding stability and binding activity”
In this section, as well as the corresponding sections in the following Materials and Methods, we give two references:
Spassov, V.Z.; Yan, L. pH-selective mutagenesis of protein-protein interfaces: In silico design of therapeutic antibodies with prolonged half-life. PROTEINS: Structure, Function, and Bioinformatics 2013, 81, 704-714.
Spassov, V.Z.; Yan, L. A fast and accurate computational approach to protein ionization. Protein Science 2008, 17, 1955-1970.
- Lines 117-122: It is not clear here, why the double mutant was chosen (see general comments).
- Figure 2: For the gel on the right it is not clear, which lanes were reduced and non-reduced. In addition, the figure Seems to be a composite of several gels. There should be a ladder added to independent gels. It is not clear, why the signals in lanes 15-18 in the second gel (R) are higher than in lanes 1-14.
- Line 139: Lanes 1-18 are mentioned, but lanes 1-23 are shown in the figure.
Response: Thank you for your suggestion. We have reorganized Figure 2, please see the details on Page 5 of our revised manuscript.
- Line 146: „could not affect its binding“. This reviewer doubts this conclusion. Looking at the figures, I see up to 1 log-fold change in EC50.
Response: It has been adjusted in the revised manuscript as follow:
Page 6, Lines: 193-197: “It also included a series of mutated antibodies in vitro (Figures S1 and S2) . The binding kinetics curves showed that both SARS-CoV-RBD and SARS-CoV-2-RBD could bind quickly and also rapidly disassociated from the antibody that conjugated to a protein chip. Not all of the modified antibodies that had been screened successfully by computational virtual calculations were found to be consistent with the experimental results.
- Figure 3 A/B/C/D: OD450 should probably read nm (nanometers) not nano molar (nM)
- Figure 3 A/B/C/D: antibody concentration in ng/ul. Greek micro needs to be replaced here.
- Figure 3 A/B/C/D: Scales are probably log scales. Either add power of 10 no the axis or log in the axis label.
- Line 153: „muTable 2.3“ The figure legend does not make sense.
Response: Thank you for your suggestion. We have reorganized Figure 3, please see the details on Page 6 of the revised verstion.
- Line 158: Reference to Figure and previous reports is missing.
Response: It has been corrected in this version. We add two references as followed:
Yuan, M.; Wu, N.C.; Zhu, X.; Lee, C.-C.D.; So, R.T.; Lv, H.; Mok, C.K.; Wilson, I.A. A highly conserved cryptic epitope in the receptor binding domains of SARS-CoV-2 and SARS-CoV. Science 2020, 368, 630-633.
Yu, W.; Wu, X.; Ren, J.; Zhang, X.; Wang, Y.; Li, C.; Xu, W.; Li, J.; Li, G.; Zheng, W.; et al. Mechanistic Insights to the Binding of Antibody CR3022 Against RBD from SARS-CoV and HCoV-19/SARS-CoV-2: A Computational Study. Comb Chem High Throughput Screen 2021, 24, 1069-1082
- Line 177-178: This sentence needs revision. Not clear, what was compared to what here.
Response: Thank you for your suggestion. It has been added in the revised manuscript as follow:
Page 7, Lines: 209-212: “CR3022 was originally found to be binding to SARS-CoV, and its affinity constant was measured to reach 2.16 nM (Figure 4A). As the affinity constant shown in Figure 4B and 4C, the affinity constant of both modified antibodies to SARS-CoV-2-RBD was improved but was not as significantly as that of SARS-CoV-2-RBD.”
- Line 186: How was significance tested? If no test was performed, „significantly“ should be replaced.
Response: Thank you for your suggestion. It has been corrected in this version and “significantly” have been removed.
- Figure 4: KD in A should be given as nM as all other KDs in this figure.
Response: It has been corrected in this version.
- Figure 4 D/E: kd is equal between D and E and does not correspond between table 2 and 4E.
Response: It has been corrected in this version.
- Figure 5B: IMGT Numbering should be given (either in addition or as replacement for PDB numbering)
Response: It has been corrected in this version.
- Line 243: „N78“ not visible from Figure 6 since different numbering was used.
Response: It has been corrected in the revised manuscript as follow: Page 11, Lines 314-315.
- Figure 6: should include IMGT numbering and label CDR1, 2 and 3
Response: Thank you for your suggestion. It has been adjusted in the revised manuscript as Figure 9.
- Lines 356-358: Should add the reference for SARS-CoV and SARS-CoV-2 RBD sequences and exactly define which amino acids were cloned for production.
Response: It has been added in the revised manuscript as follow:
Page 4, Lines: 426-429: “The SARS-CoV and SARS-CoV-2 RBD amino acid sequences were obtained from GeneBank with its corresponding ID numbers: NP_828851.1 (SARS-CoV-2 RBD) and YP_009724390 (SARS-CoV-RBD). “
Line 448: How was significance tested?
Response: Thank you for your suggestion. We confirm that this part of the text is not suitable here, we have removed the relevant part of the description, and we have discussed in detail the reasons for choosing double-site mutations in the previous conclusion section in the revised manuscript as follow: Page 4, Lines 125-137.
- Line 466: This sentence seems to make no sense here.
Response: It has been corrected in the revised manuscript as follow:
Page 17, Lines 563-566: “The affinity of an antibody is not directly equivalent to in vitro neutralizing ability. Likewise, neutralizing capacity in vitro is not directly equivalent to protective capacity in vivo. Therefore, the improvement of the neutralizing activity of CR3022 antibody still needs to be explored. ”

Reviewer 2 Report
To develop a neutralizing antibody against SARS-COV-2, the SARS-CoV antibody CR3022 was affinity matured by the structure-based molecular simulations in the current study. The residues in the CDRs of CR3022 antibody were mutated and the affinity was measured as low as 1~7nm, which is close to the affinity of the published SARS-CoV-2 antibodies.
The paper showed a new way to develop neutralizing antibody against unkown pathogens in short time. However, I would suggest that the antibody should be tested by the psuedovirus in vitro.
Author Response
Response for Reviewer 2:
Comment #1: The paper showed a new way to develop neutralizing antibody against unkown pathogens in short time. However, I would suggest that the antibody should be tested by the psuedovirus in vitro.
Response: Thanks for the suggestion. In this version, the corresponding neutralization experiments have been added in this revised manuscript. We have made changes marked up using the“Text marked in yellow color” in the revised manuscript as section 2.5 and section 3.7.
Corresponding neutralization experiments have been added to this revised manuscript, and although the neutralization titer has not been significantly improved, the ability to inhibit the virus has been improved.
At the same time, we also noticed that there are reports that the CR3022 antibody itself does not have neutralizing activity against SARS-CoV-2 in vitro. However, it cannot be deduced from this that it has no protective effect in vivo.
The above conclusion can be confirmed from a published Science article (Yuan, M. et al. Science 368, 630-633) which mentioned in the article: In the in vitro micro-neutralization experiment, the concentration of CR3022 reached 400 μg/mL, and the effect of neutralizing the SARS-CoV-2 virus was not seen, but it cannot be said that CR3022 cannot protect individuals against SARS-CoV-2 in vivo.
Due to the specificity of the CR3022 binding epitope which does not overlap with the RBD-ACE2 binding site, may lead to the generation of a "breathing pattern". This situation is similar to that antibodies targeting the surface epitope of the HA trimer of influenza A virus do not neutralize in vitro, but protect individuals against influenza A virus attack in vivo. (Bangaru, S. et al. Cell 177, 1136-1152 e1118; Watanabe, A. et al. Cell 177, 1124-1135 e1116; Bajic, G. et al. Cell host & microbe 25, 827-835 e826)
Therefore neutralization experiments in vitro cytology are not directly equivalent to the protective effect of antibody drugs in vivo,the purpose of our article is to improve affinity, but affinity is not equal to neutralization ability in vitro, but it does not mean that there is no protective effect in vivo.
In conclusion, the combination of computational and experimental approaches in this article can provide insights for developing as broadly neutralizing antibody drugs against SARS-CoV-2, which is the main purpose and value of the article.

Round 2
Reviewer 1 Report
This reviewer highly appreciates that the authors addressed each of the questions and concerns. In particular, the authors provided the requested auto-reactivity and neutralization assays. Overall, this article has some potential.
However, this reviewer would have appreciated a more careful revision by the authors. There are still numerous errors, including wrong citations (e.g., CR3022 has been first described in PLOS Medicine https://doi.org/10.1371/journal.pmed.0030237 and not in the cited J Virol 2005 paper https://doi.org/10.1128/JVI.79.3.1635-1644.2005), inconsistencies in number presentations (see line 193ff, '14.6x10-9' vs. '1.17E+06'), wrong words which might probably have resulted from auto-correction (e.g. line 24, 'CDR' stands for 'complementarity-determining region', not 'complemented-determined region'), as well as typos and missing words (e.g. line 22 "... to SARS-CoV-2 by in vitro affinity", 'maturation' is missing). Importantly, there are also misleading conclusions from the novel neutralization experiment (Figure 6). The data looks noisy and it seems that several CR3022 data points are single values. As a consequence, the fits for CR3022 and S103Y-S33R look inappropriate. This reviewer can not interpret any difference between CR3022 and the two mutants from this data and would not conclude an improvement of any inhibitory effects here.
In summary, this reviewer likes the applied combination of in silico predictions to improve affinities and the experimental validation of these predictions. He does agree that the authors present a comprehensive approach to generate and improve cross-reactive antibodies. However, neutralization was not improved and this reviewer does therefore not agree on the conclusions that the manuscript provides insights for developing broadly-neutralizing antibodies against SARS-CoV-2 (lines 37-40, 561-63, 570-571). Since the presented data is not self-explaining for this conclusion, the authors would still need to elaborate on how exactly the presented approach (i.e. improvement of affinity) can be exploited to generate broadly-neutralizing antibodies, or concentrate on cross-reactivity only.
Unfortunately, due to the erroneous presentation of the manuscript, the questionable interpretation of the neutralization data, and the inconclusive link between the affinity maturation approach and neutralizing antibodies, this reviewer would not consider the current version of this manuscript for publication in Viruses.